# Osteopathic Treatment and Evaluation in the Clinical Setting of Childhood Hematological Malignancies

**DOI:** 10.3390/cancers13246321

**Published:** 2021-12-16

**Authors:** Monica Barbieri, William Zardo, Chiara Frittoli, Clara Rivolta, Valeria Valdata, Federico Bouquin, Greta Passignani, Alberto Maggiani, Momcilo Jankovic, Andrea Cossio, Andrea Biondi, Adriana Balduzzi, Francesca Lanfranconi

**Affiliations:** 1Maria Letizia Verga Center, Department of Pediatrics, University of Milano-Bicocca, ASST Monza/MBBM Foundation, 20900 Monza, Italy; barbieri.mnc@gmail.com (M.B.); williamzardo69@gmail.com (W.Z.); chiarafrittoli88@gmail.com (C.F.); rivoltaosteopata@libero.it (C.R.); valeriavaldata@yahoo.it (V.V.); federico.bouquin@gmail.com (F.B.); gretapassignani@gmail.com (G.P.); momcilo@libero.it (M.J.); abiondi.unimib@gmail.com (A.B.); abalduzzi@fondazionembbm.it (A.B.); 2Division of Research, Italian Academy of Osteopathic Medicine (AIMO), 21047 Saronno, Italy; alberto.maggiani@aimoedu.it; 3Department of Orthopedics, University of Milano-Bicocca, ASST Monza, 20900 Monza, Italy; cossio.ortopedia@gmail.com

**Keywords:** hematology, pediatric, osteopathy, precision-based exercise, leukemia, lymphoma, hematopoietic stem cell transplantation

## Abstract

**Simple Summary:**

Children: adolescents, and young adults who are affected by hematological malignancies and who are undergoing intensive phases of cancer treatment including hematopoietic stem cell transplantation, experience diminished functional ability. This study was aimed at assessing if osteopathic treatment and evaluation can be used when an 11-week precision-based exercise program is run inside the hospital. Our results support that osteopathy plus precision-based intervention could be a desirable support in the clinical *prise en charge* of these children and adolescents. Osteopathy is a safe method for the evaluation of clinical conditions and requires strong multidisciplinary synergy between pediatricians and exercise physiologists.

**Abstract:**

Children: adolescents, and young who are adults affected with hematological malignancies (CAYA-H) and who are undergoing intensive phases of cancer treatment, including hematopoietic stem cell transplantation (HSCT), experience diminished functional ability. This study was aimed at assessing the feasibility, efficacy, safety, and satisfaction of an osteopathic intervention in CAYA-H attending an 11-week precision-based exercise program (PEx). All of the participants were given 4–10 treatments according to the prescription ordered by the sports medicine doctor in charge of the PEx, and the following outcomes were assessed: (1) spinal column range of motion (ROM) by palpation; (2) lower and upper limb joints ROM by a goniometer; (3) orthostatic posture by plumb line assessment; (4) chest and abdomen mobility by inspection and palpation; (5) cranial-sacral rhythmic impulse (CRI) by palpation; and (6) adverse effects. Goal attainment scaling (GAS) was used to identify the accomplishment of a desired clinical result. Moreover, HSCT patients who were affected with graft-versus-host disease and/or osteonecrosis had their joints assessed in terms of ROM as tools to monitor the effectiveness of immunosuppressive treatment. A total of 231 CAYA-H were identified, and 104 participated in the study (age 10.66 ± 4.51 yrs; 43% F). PEx plus osteopathy reached positive GAS scores by improving the ROMs of the spinal column and/or limbs (81% and 78%, respectively), chest and abdomen mobility (82%), and CRI (76%). Only minor reversible adverse effects were noticed during the study. Together, our data seem to initiate a new course where osteopathy could be useful in evaluating structural edges due to the clinical history of each CAYA-H. Given the contributions that were obtained by the GAS scores, osteopathic treatment seems to reveal interesting potential that can be targeted in the future.

## 1. Introduction

According to the European Society of Paediatric Oncology’s (SIOPE) perspective, there are nearly 500,000 European children, adolescents, and young adults with hematological malignancies (CAYA-H) today. In 2030, there will be around 750,000 CAYA-H survivors: two-thirds of them will have some long-term effects and/or sequelae due to treatment and/or complications [1,2]. Cardiomyopathy, pulmonary diseases, myopathy, osteoporosis, and osteonecrosis and motor nervous function impairments are some of the consequences of cancer treatment that can jeopardize the possibility of performing physical activity [3,4,5,6]. Pain, fatigue, anxiety, and fear are the final causes of the reduced exercise tolerance in CAYA-H [7,8]. Exercise capacity is reduced at the beginning of treatment, after being off-therapy for one year, and even after 15 years of follow-up [9,10,11,12]. Thus, adapted precision exercise-based training programs (PEx) are an emerging therapeutic option as an adjuvant treatment in pediatric oncology [13,14,15,16,17]. Although a recent review suggested that the effects of PEx for childhood cancer participants are not yet convincing due to the small numbers of participants and insufficient study designs, the first results show some positive effects on physical fitness in the intervention groups compared to the control groups [18]. There were positive intervention effects in terms of body composition, flexibility, cardiorespiratory fitness, muscle strength, and health-related quality of life (cancer-related items). In the last 2 years, new studies on large cohorts of CAYA-H showed how supervised in-hospital PEx for children with cancer are safe and play a cardio-protective role as well as improve exercise capacity [19,20].

A totally new field of research, one that has been neglected in the scientific literature dedicated to PEx, concerns the possible effect of manual treatment added to exercise in CAYA-H. Complementary therapies, such as osteopathy, can help to manage the exasperating cluster of signs and symptoms, relieving the physical and emotional burden of CAYA-H and their families [21]. As confirmed by the WHO in 2019, osteopathic medicine relies on manual contact for diagnosis and treatment [22]. Osteopathic treatment (OT) can help to manage pain relief and quality of life (QoL) in patients with cancer and in very frail patients with neurodegenerative diseases [23,24], and it has also been shown to improve functional status in acute and chronic pain of nociceptive or neuropathic origin in different clinical conditions [25,26,27]. Several studies have demonstrated a possible anti-inflammatory effect of OT, showing a reduction of inflammatory cytokines (IL-6, IL-12), substance P, and TNFα [28,29].

Although PEx are known to impact exercise tolerance in CAYA-H, as reported in previous studies, the effect of exercise plus OT has never been tested. In order to deal with CAYA-H in a patient-centered way and to allow our study to integrate the outcomes of patients with very heterogeneous symptoms, we arbitrarily decided to use a broad-spectrum tool to quantify the treatment benefits: goal attainment scaling (GAS). GAS is an assessment instrument that can be used to evaluate interventions based on individual, patient-specific goals [30,31]. This is of particular importance in clinical studies in diseases where due to the diversity of symptoms, it is especially difficult to perform an inferential analysis based on a single quantified outcome, i.e., a single standardized endpoint. The symptoms and signs of CAYA-H during cancer treatment are diverse and range from minor inconveniences that compromise quality of life to being life-threatening as a result of side effects that are related to cancer treatment. The most common symptoms and signs are joint and skeletal muscle pain at rest and during exercise, headache, flushing, itching, abdominal pain, nausea, vomiting, heartburn, tachycardia, odynophagia (difficult swallowing), sialorrhea, dyspnea, and hyper or hypotension. Osteopathic evaluation (OE) seems to be an appealing support for the evaluation of diminished joint range of motion (ROM) and postural unbalance due to problems originating from various parts of the skeletal muscle system (structural function). OE could also eventually support the pre-clinical diagnosis of symptoms and signs that are characteristic of CAYA-H in their intensive phases of cancer treatment, including a differential diagnosis within the specific thorax and abdominal organ disorders that manifest during inspection and palpation (visceral function).

This study aimed to assess the safety, feasibility, efficacy, and satisfaction of OT in CAYA-H during their cancer treatment plus an 11-week precision-based exercise program. Additionally, OE was tested as a possible complementary procedure to safely evaluate clinical conditions in CAYA-H before and after the exercise program. Specifically, using a GAS, the following indicators were tested: (1) the ROM of the cervical-dorsal-lumbar spine; (2) the ROM of the upper and lower limb joints; (3) the orthostatic posture; (4) the visceral function of the thorax and abdomen; (5) the cranial and sacral rhythmic impulse (CRI); and (6) the possible adverse effects of exercise plus OT.

## 2. Materials and Methods

Every attempt was made to avoid unnecessary discomfort and disturbance to the CAYA-H and their parents according to the ERICE statement about the cure and care of long-term survivors of childhood cancer [32]. All of the children and their parents provided informed assent and written consent to participate in the study, and it was approved by the University of Milano Bicocca ethics committee (registered number 2017/284). Personal data were treated in compliance with the European standard principles of confidentiality (n. 2016/679). The protocol was registered to ClinicalTrials.gov PRS (n. NCT04090268).

### 2.1. Precision-Based Exercise Protocols and OT

Eligible CAYA-H were those treated for any hematological malignancy (including relapses) at the Maria Letizia Verga center (Monza—Italy) from April 2017 to December 2019, with the participants being aged 3 to 19 years old. Another criterion was that each CAYA-H agreed to participate in a round of an 11-week combined training program (endurance, resistance, balance, and flexibility) three times per week that was individually targeted, with workloads ranging from moderate to intense exercises, depending on the daily clinical conditions. Each participant was informed by the referring pediatric hematologist of the possibility of participating in the research protocol. Subsequently, a sports medical doctor evaluated the possible risks of performing precision training or OT. The CAYA-H were allocated in two precision exercise protocols and OT depending on their clinical history and according to the intensity of their chemotherapy protocol (See Appendix B).

The osteopaths received a 4-month ad hoc training period, which was provided by the sports medicine doctors. They treated the CAYA-H during the 11 weeks of PEx. The OT techniques that were used during the treatments were focused on recovery from structural or visceral cancer-related consequences of chemotherapy, immobilization, and/or exercise training (e.g., vertebral crush fracture). All the participants showing joint pain and/or ROM limitations, myalgia, acute abdominal or lung infection or toxicity (especially transplant recipients), sleep disorders, nausea, headache, chronic chest wall restriction, and active and/or outcomes of osteonecrosis underwent 4–10 OTs, as per the prescription by the referring physician (see Appendix B).

### 2.2. Osteopathic Evaluation

CAYA-H assessment was performed before (T0) and after (T1) the 11 weeks of PEx and OT and at any time when an intercurrent condition raised. The OE was assessed and noted using a medical record that had been divided in sections that evaluated the spinal column and the ROM of the lower and upper limb joints, the orthostatic posture [33], a clinical examination of the chest and abdomen, and a CRI [34,35] (see Appendix B).

Every OE and/or exercise training session was reported on a daily basis for each patient and was included in their clinical report. Any relevant adverse effects were considered as possible outcomes of interest and were clearly identified. Before the onset of the study, we arbitrarily decided to set a list of possible adverse effects to be followed up with in the 2 days after the OT session: neurological (vertigo, nausea, headache), musculoskeletal (joint or muscular pain), gastrointestinal or urogenital (diarrhea, abdominal pain), and subjective discomfort during the treatment.

When interpreting outcomes with a GAS endpoint, it is essential that the chosen goals are reported upon in a similar way as baseline variables are traditionally reported upon in randomized controlled trials; to ensure that this was the case in the present study, we ran a 1-year extensive investigation before starting the OE. Thus, we were able to categorize the goals based on the most common parameters that we found to be recurrent and noticeably impaired (e.g., dorsal and lumbar dysfunctions of the column due to the presence of a central venous catheter inserted at the second rib level or back pain as the principal structure enduring painful stiffness when long bed-rest periods occur). Finally, we built a basal OE that was used to establish one or more rehabilitative goals by a GAS (see Appendix B). None of the osteopaths who participated in the final OE had any prior knowledge of the initial goals that had been established [36].

### 2.3. Evaluation of the Satisfaction of the OT Intervention

The visual analogue scale (VAS) method was used to evaluate the satisfaction of both participants and their parents after 11 weeks of OT. The VAS is a patient-reported scale that ranges from 0 to 10 (where 0 is not satisfied and 10 is completely satisfied) [37].

### 2.4. Study Design and Statistical Analysis

This was a clinical, analytic, observational, cohort study, where the magnitude of the intervention was also evaluated longitudinally. Values were expressed as mean (±standard deviation). Sample size calculation determined that a sample of 15 participants would be adequate to detect a difference of 25% in the exercise capacity between the participants and the healthy children and was determined to have a power of 0.80 (α = 0.05). The column, subtype box and violin, and violin plot graph families were used. All of the statistical analyses were performed using a commercially available software package (Prism 8.0: GraphPad, La Jolla, CA, USA).

## 3. Results

Two hundred thirty-one CAYA-H were eligible for participation in the study (age 10.6 ± 4.5 yrs; 43.0% F). Eighty-one (35%) CAYA-H did not participate in the study, the main reason for which was due to logistical reasons (out of province or region). A small number of families refused to attend the PEx and OT sessions because of a lack of interest in participating in the research project (5%). Finally 150 CAYA-H were recruited and 24 (16%) dropped out (<15% of adherence to the PEx sessions) due to lack of compliance such as living too far away from the hospital or a busy familiar schedule when young siblings were in the same familiar nucleus or in cases of death (four deceased CAYA-H).

OT was also performed in severe clinical conditions, when the active PEx was unbearable. In fact, 22 CAYA-H received just OT: because their clinical conditions were substantially different from the rest of the participants enrolled, their data are not presented in this paper.

One-hundred four CAYA-H attended more than 16% of the PEx and more than 3 OT sessions. Only the CAYA-H who attended more than 30% of PEx and more than 4 OT sessions were considered for GAS scores at T1 because the effect of the intervention is only reliable when there is medium or high adherence to the training sessions [20]. Patients with adherence to the PEx that was lower than 29% were considered as a low adherence group, and their data are not presented in this paper. The flow diagram of participants is visible as a Appendix A as per the “Strengthening the Reporting of Observational Studies in Epidemiology (STROBE) Statement: guidelines for reporting observational studies”.

Table 1 shows the characteristics of CAYA-H accordingly with their adherence to the PEx and OT and divided by cohorts of age, sex, malignancy type, treatment protocol, HSCT percentage for each malignancy, deaths.

Thirty-three participants (32%) were enrolled for more than one round because their functional capacity and exercise tolerance were far from the established goals. Osteonecrosis (26%) and/or GvHD (8%) at the limb joints were the recurrent conditions in this cohort of CAYA-H: the main part attende as pre or post intercourse for osteonecrosis and/or joints’ GvHD.

The more relevant structural and visceral dysfunctions at both T0 and T1 were the reduced extension of the dorsal spine (59% and 47%, respectively); reduced extension of the sacral spine (42% and 27%, respectively); reduced flexion of the ankle (20% and 12%, respectively) and reduced internal rotation of the hip (22% and 12%, respectively); unbalanced forward posture (8% and 7%, respectively); impaired diaphragmatic expansion capacity at both T0 and T1 (61% and 38%, respectively); and reduced CRI (85% and 63%, respectively).

In Figure 1, the GAS scores of the spinal column functions that were obtained at T1 are shown. A GAS score of “0” was achieved in all of the functional parameters in 41% of the OEs. Better outcomes than expected (score +1 and +2) were also represented (33% and 7%, respectively). Scores that did not change (−1) or that worsened (−2) were in the minority (9% and 10%, respectively). The spinal functions where the lower scores were mainly represented were the reduced extension in the lumbar, dorsal, and cervical districts.

In Figure 2, the GAS scores for different leg functions are shown. In 40% of cases, the GAS reached a score of “0” (ROM increased by 3°). Better outcomes than expected (score +1, ROM increased by 4–6° and +2, ROM increased more than 6°) were also represented (31% and 7%, respectively). No change (−1) or worsening scores (−2) were a minority (9% and 13%, respectively). The structural functions where the lower scores were mainly represented were the external and internal rotation of the hip and the flexion of the ankle.

In Figure 3A, the GAS scores for the CRI sacral functions are shown. In 43% of cases, a GAS score of “0” was obtained in all of the parameters. Better outcomes than expected were represented (28% and 5%, respectively). No change or worsening scores were 12% and 13%, respectively. The function where the lower scores were mainly represented was sacral amplitude.

In Figure 3B, the GAS scores of the CRI cranial functions are shown. In 42% of cases, a GAS score of “0” was obtained for all of the parameters. Better outcomes than expected (score +1 and +2) were represented (30% and 5%, respectively). No change or worsening scores were represented in 14% and 9% of cases, respectively. The function where the lower scores were mainly represented was cranial amplitude.

The most prevalent abdominal visceral dysfunctions were at the right and left iliac regions and at the left hypochondriac and lumbar regions, where pain during superficial and deep palpation and/or altered bowel movements were observed. When a possible clinical red flag was found, the pediatrician in charge of the patient was promptly consulted, and the finding reported on the clinical report created by the sports medical doctor. In 68% of cases a GAS score of “0” was obtained, while no changes were evident in 32%. The most common chest dysfunctions were related to the diaphragm and 1° and 2° rib mobility. In 82% of cases, a GAS score of “0” was obtained, while in 18%, no changes were evident.

Only minor adverse effects were reported in a few cases after the OT sessions (1.7%): three patients stated moderately increased joint pain the day after the treatment, and one reported increased nausea. Medical doctors promptly evaluated the patients, and a re-tailored OT and/or medication was prescribed in order to alleviate the symptoms. One adolescent refused to be treated by certain osteopaths but agreed to be followed up with by others.

Overall, the average VAS score regarding the PEx showed considerable satisfaction by both the CAYA-H and their parents (9.30 ± 1.25 and 9.64 ± 0.60, respectively). When the OT intervention was considered on its own, the average VAS score was also high (9.06 ± 1.25 and 9.47 ± 1.25, respectively).

## 4. Discussion

This study attempted to prove the feasibility and safety of osteopathy in a pediatric hematologist-oncological cohort that included very frail patients who enrolled in a precision exercise-based training protocol. A well-coordinated and collaborative multi-disciplinary prise en charge of patients with CAYA-H allowed for the inclusion of healthcare professionals who were new to the paediatric hemato-oncological setting, such as sports medicine doctors, osteopaths, exercise physiologists, and other specialists (orthopedics, physiatrists). The shared OE and the tailored OT plan allowed for the expression of the rehabilitative potential of manual therapy integrated with the PEx. Osteopathy has been previously used in adult patients with cancer as compassionate therapy [38,39,40]. CAYA-H sometimes have challenging clinical histories due to miscellaneous and sometimes simultaneous causes of both structural and visceral origin. A characteristic point of structural dysfunction was seen at the cervical-dorsal level, and despite setting goals for better mobility, these goals were not able to be reached in all cases. of the main reason behind reduced cervical-dorsal mobility is the presence of a central venous catheter (CVC) or a peripherally inserted central catheter (primarily positioned in one arm). As reported by Brisson et al., reduced spine mobility can be a complication of CVC insertion in children under general anesthesia [41]. Apart from the possible surgical side-effects, patients are prone to adopting a typically defensive posture to protect the devices from detaching during their daily life activities.

OE was an important tool for patient clinical assessment in order to identify where the thoraco-pulmonary areas lacked excursion, which was investigated both passively and actively. The data in this study showed that in CAYA-H, the diaphragmatic medial and lateral pillars had a reduced inspiration capacity, especially when the clinical history involved long periods of bed rest. OT, which acted as a support to the PEx, was prescribed to improve and maintain the respiratory muscles elasticity, by passively elongating the muscle fibers in the diaphragm pillars to counteract the stiffness that had been acquired during the patient’s clinical history.

From a visceral point of view, pulmonary toxicity is a common long-term complication that results from infections as well as from exposure to certain anticancer therapies in childhood and can vary from subclinical to life threatening events [42]. Certainly, the iatrogenic consequences of a reduced thoraco-pulmonary compliance in CAYA-H have been attributed to radiotherapy, anticancer treatment, and eventually to GvHD in HSCT recipients [43,44,45,46]. The OE evaluated the progress that was being made during the PEx training that was aimed at improving the mechanical capacity of the thoraco-pulmonary system, and OT supported efforts to counteract the stiffness of the system.

The lumbar spine is the principal structure in which painful stiffness endures after long periods of bed rest, and substantial skeletal muscle oxidative capacity is lost in 30 days [47]. On the other hand, growing adolescents exhibit a peculiar posture that characterized by the shortening and spasming of the psoas muscles, the shortening of the abdominal muscles, and/or increased lumbar lordosis and anterior tilt of the pelvis [48]. The primary reason for OT support during the 11 weeks of PEx training was to improve the core muscles strength (pelvic floor, abdominal wall, back, and diaphragm muscles) parameters, the strengthening of which would consequently improve lumbar mobility. OE permitted the lumbar spine mobility condition that is present in CAYA-H after long periods of bed rest (i.e., mobility severely reduced) to be differentiated when compared to CAYA-H who were more active (i.e., mobility closer or equal to the physiological function).

From a visceral perspective, some of the CAYA-H with back pain who had sacro-lumbar vertebral structures demonstrating limited mobility showed associated urogenital or intestinal problems. Song et al. found that in adolescents, the musculoskeletal system can often contribute to chronic pelvic pain, a dysfunction that may develop as a response to gynecological or non-gynecological problems [48]. The OE of these CAYA-H helped to support pediatricians make a differential diagnosis between primary structural system issues (myofascial pain) vs. possible diseases of the reproductive organs, bowel syndromes, or urinary problems [49]. An example of this was a case of severe acute back pain that resulted in a walking impairment and that was considered to be of spinal origin as it did not demonstrate any signs of abdominal tenderness. Because the OE was strongly indicative of the pain possibly being of a pelvic organ origin, a CT scan was requested after a consultation between the osteopath, the sports medicine doctor, and the pediatrician. Finally, the pain was determined to be the result of a supraelevator rectal abscess. Introducing OT in CAYA-H adds novel insights and convenient, non-invasive means to investigate pelvic pain, especially considering the reluctance of young patients when these anatomical parts are involved.

In CAYA-H, specific joints have long-term reduced function., e.g., ankle flexion and/or lower limb muscle strength, due to specific drugs such as vincristine and corticosteroids [20,50,51,52]. Our OE confirmed all of the signs that have been previously reported, according to the phase treatment, especially in the intensive phases.

Osteotoxic chemotherapy, prolonged treatment with steroids, poor nutrition, vitamin D deficiency, and poor muscle mass are recognized risk factors that contribute to bone pathology during and after cancer therapy, resulting in negative skeletal configurations, osteoporosis, long bone and vertebral fractures, and osteonecrosis [53]. The latter is a serious and debilitating complication in CAYA-H, of which incidence varies according to diagnosis, age, sex, and treatment protocols, with 5 to over 50% of the pediatric patients experiencing these side effects [54,55]. Osteonecrosis is an elusive condition with a challenging diagnosis due to a heterogeneous manifestation, especially when there is already severe joint damage that is present. OE allowed the clinical follow up of active osteonecrosis at hip, knee, and ankle sites, locations where the worst sequelae occur, by identifying which CAYA-H were at risk for significant progression and required an intense orthopedic evaluation.

Another peculiar complication that involves the upper and lower limb joints is chronic GvHD, a problem that affects 5–30% of allogenic HCST recipients. GvHD causes different degrees of severe extremity contractures from skin and fascial inflammation, poor aerobic capacity from pulmonary involvement and deconditioning, dry mucous membranes (e.g., eyes and mouth), oral ulcers, and edema from hepatic congestion [56,57,58]. In our study CAYA-H with GvHD were included and OE specifically measured the benefit of each line of treatment, especially when joints were involved. PEx plus OT seemed to impact the decreased ROM of the upper and lower limbs once anti-rejection medications reached their maximum therapeutic potential. Certain CAYA-H were able to improve their ROM by a few degrees, bring them closer to the full physiological ROM, representing a considerable step in a better QoL.

A literature review revealed that a CRI of cranio-sacral function in CAYA-H has never been investigated. A correspondence with the clinical status of our patients was noticed: very frail patients, including the four children who were treated at the end of their lives, showed reduced expression of all of the CRI parameters. On the other hand, children who were more physically active, i.e., in the medium intensive phases of treatment or after therapy discontinuation, showed a more appreciable CRI.

In CAYA-H, the gut microbiota is modified after chemotherapy or radiotherapy administration, leading to gastro-intestinal side effects such as nausea, vomiting, diarrhea, abdominal distention, and constipation [59]. Mucositis is the main cause of abdominalgia and compromises intestinal function, which affects nutritional status and QoL, eventually affecting survival [60]. Nausea and vomiting are the most common side effects of chemotherapy [61]. Manual treatment on chemotherapy-induced nausea and vomiting in pediatric cancer have been used successfully to eliminate nausea or to reduce the severity, length, and frequency of nausea and vomiting [61,62]. The potential auxiliary outcomes of manual treatment effects include increased vagal activity and decreased stress hormones [63]. Applying OT in CAYA-H improved our knowledge of the potential of manual treatment in easing these vicious symptoms, and it was possible to monitor gastro-intestinal function by OE. Allowing parental caregivers to actively participate in reducing their child’s discomfort by allowing them to repeat some manual maneuvers can reduce their anxiety and can create a synergic alliance with the health figures.

A lot of time was spent in training each osteopath to report every OE that was performed on each patient on a daily basis. We considered relevant adverse effects as possible outcomes of interest because we are aware of the fact that adverse effects data are often handled with less rigour than the primary beneficial outcomes of a study [63]. Because only reversible minor adverse effects were noted during this study, we considered osteopathy as being successfully introduced in a complex clinical setting. When a possible pre-clinical sign or symptom was found, we promptly referred it to the pediatrician who was in charge of the patient.

The limitations of this study, which are in accordance with the “imaginative solution” proposed by Zucchetti et al. [64], are the lack of possible randomization criteria for the participants in intervention and control groups. These authors also suggest the evaluation of common types of side effects such as fatigue, which we assessed only indirectly, to be among the reasons that justify a lack of adherence to PEx. We are also definitely aware of the fact that a randomized control trial would have provided the highest evidence level for this experimental intervention. We evaluated a CTRL group of healthy children and adolescents, and their functional performance was assessed in order to verify how far our CAYA-H were from their peers in terms of exercise capacity [20]. In terms of having a CTRL group of CAYA-H, upon a careful evaluation, two true homogenous groups could not be established within these 2.5 years in the experimental setting. We would have to consider the different diseases (leukemia type, lymphoma type), the different treatment protocols, the standard vs. high-risk line of treatments, sex and age, clinical complications such as secondary lung infections or post-transplant GvHD. We accept that our study has a lower ranking in the hierarchy of evidence, as losing the power of randomization causes the study to be more susceptible to bias and confounding. However, the large cohort of patients and the GAS score along with average positive goal achievement made it fair to conclude that the intervention might be responsible for the improvements that were seen in the present study. We hope that more researchers will add evidence proving the benefit of the use of manual treatment in children with cancer settings.

## 5. Conclusions

When following medical prescriptions, osteopathic intervention appeared to be safe and effective in pediatric hemato-oncological patients, who expressed a high level of appreciation. This study suggests that osteopathic evaluation and treatment can be integrated to precision-based exercise programs, even when children, adolescents, and young adults are undergoing the most intensive phases of the cancer treatment.

## Figures and Tables

**Figure 1 cancers-13-06321-f001:**
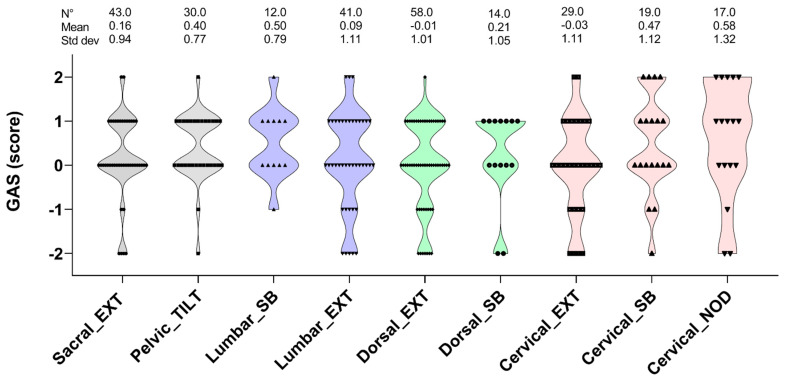
The violin plot graph shows the goal attainment scaling (GAS) scores of the spinal column mobility divided into sacral, pelvic, lumbar, dorsal, and cervical districts after 11 weeks of precision-based exercise plus osteopathic treatment. Only children, adolescents, and young adults with medium or high adherence to the training sessions and osteopathic treatments were considered (see the method section for further details). EXT, extension; SB, side bending, NOD, nodding.

**Figure 2 cancers-13-06321-f002:**
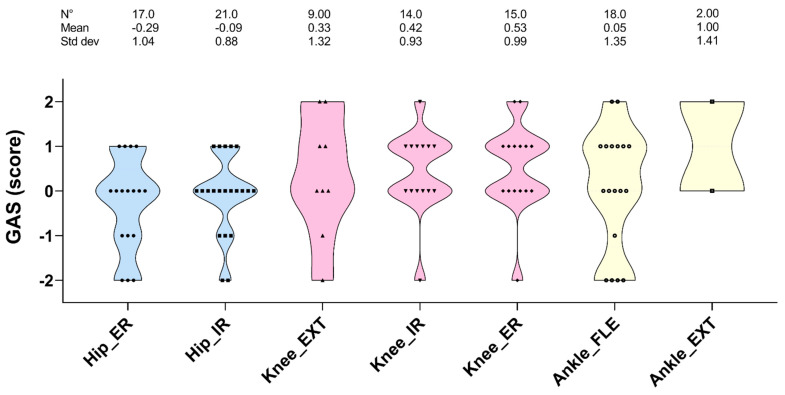
The violin plot graph shows the goal attainment scaling (GAS) score for the range of motion in the lower limbs, divided in hip, knee, and ankle districts after 11 weeks of precision-based exercise plus osteopathic treatment. Only children, adolescents, and young adults with medium or high adherence to the training sessions and osteopathic treatments were considered (see the method section for further details). ER and IR, external and internal rotation; EXT extension; FLE, flexion.

**Figure 3 cancers-13-06321-f003:**
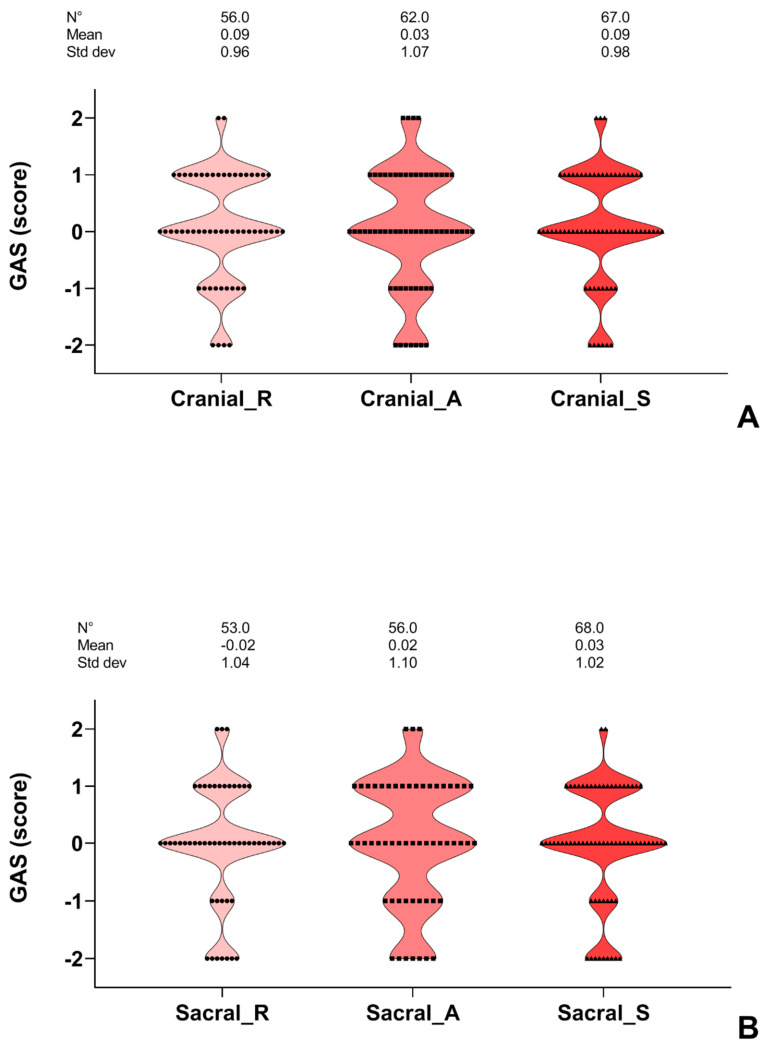
(**A**,**B**) The violin plot graph shows the goal attainment scaling (GAS) scores of the cranial and sacral impulse (CRI) districts. Only children, adolescents, and young adults with medium or high adherence to the training sessions and osteopathic treatments were considered (see the method section for further details). R, rhythm; A, amplitude; S, strength.

**Table 1 cancers-13-06321-t001:** Clinical characteristics of CAYA-H divided into high, medium, or low adherence groups (HAd, MAd, LAd, respectively). ALL: acute lymphoblastic leukemia; AML: acute myeloid leukemia; HL: Hodgkin lymphoma; NHL: nonHodgkin HSCT: hematopoietic stem cell transplant.

	Had—MAd	LAd
Patients (n)	75	29
Age (yrs)	11.08 ± 4.46	10.00 ± 4.30
Age (yrs, range)	3–22	3–18
Sex (% Female)	54.50	46.00
Aged 3 < x < 11 yrs (%)	50.13	50.00
Aged 12 < x < 22 yrs (%)	49.87	50.00
*Treatment protocol*		
ALL (%)	59.70	62.29
HSCT (%)	27.78	28.95
*Treatment protocol*		
AML (%)	10.50	9.84
HSCT (%)	57.79	66.60
*Treatment protocol*		
HL (%)	13.28	9.84
HSCT (%)	18.83	50.00
*Treatment protocol*		
NHL (%)	6.18	4.91
HSCT (%)	18.39	100.00
*Treatment protocol*		
Non onco (%)	10.35	13.11
HSCT (%)	71.60	75.00
>then 1 round (%)	32	

## Data Availability

The data presented in this study are available on sensible request from the corresponding author. The data are not publicly available due to sensible information regarding the clinical status of underage patients.

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
