# Peer review of "Osteopathic Treatment and Evaluation in the Clinical Setting of Childhood Hematological Malignancies"

_cancers, 2021, doi:10.3390/cancers13246321_

Round 1

Reviewer 1 Report

The authors full replied to the majority of criticism.

This study confirms the role of exercise in the long-term wellness of paediatric patients undergoing chemotherapy /HSCT treatment.

The role of of OT needs more details.

Author Response

We thank the Reviewer for her/his scrutinized revision. We hope that more studies will add evidence to the use of manual treatment in children with cancer settings. We added this sentence to our discussion.

Reviewer 2 Report

Accepted in present form

Author Response

We thank the Reviewer for her/his scrutinized revision. We hope that more studies will add evidence to the use of manual treatment in children with cancer settings. We added this sentence to our discussion.

This manuscript is a resubmission of an earlier submission. The following is a list of the peer review reports and author responses from that submission.

Round 1

Reviewer 1 Report

The text by Barbieri et al is interesting because it attempts to explain and introduce the osteopathy technique in patients suffering from haematological diseases in the active phase of treatment. The authors want to show the results of a study in which osteopathy technique is associated with physical exercise in a large number of patients. The authors report a limited number of dropouts (16%). The authors subsequently measure the observed differences between T0 and T1.

Limitations of the study:

The main limitation of the study is that there is no control group for which it is not possible to document the real role of these two physical techniques. Control of individual patients towards themselves at T0 is not convincing. The authors should indicate the characteristics of the patients upon entry into the study (disease / gender / age) and subsequently analyze the data according to the groups. It seems likely that younger children than young adults may biologically present a category more easily to recovery. 

Author Response

We thank the reviewer for her/his careful evaluation that helped us in improving  our manuscript. In the attached file there are our point by point answers.

Reviewer 2 Report

In this manuscript Barbieri and colleagues describe the results of a prospective single arm interventional study aimed to test the safety, the feasibility and the efficacy of osteopathic treatment in a cohort of 231 pediatric patients undergoing hematopoietic stem cell transplantation.

The topic is really interesting and of potential interest for the readers, and the literature about it is limited so far.

The results are adequately presented and discussed.

As minor comments I would better specify in the introduction that while other types of physical supportive therapy have been investigated in this population (i.e. Bram et al Cochrane Database 2013) the data about osteopathic treatment are scarce. Moreover in the discussion I'd introduce a comment about the results of the present study and those of the studies reviewed by Zucchetti et al in Pediatr Blood Cancer 2018.

Author Response

(The authors gave the same response as above.)

Reviewer 3 Report

Barbieri et al present a study on osteopathic treatment in survivors of childhood haematological malignancies. Although this study highlights an interesting topic and supportive treatments are very valuable for this severely affected patient group, there are some major content and methodological weaknesses:

  • Without a control group, it is difficult to assess the impact of the intervention as well as satisfaction with the program in this cohort as it might also be a consequence of more time and attention. Also, the natural course of physical limitations after completion of oncological therapy is not presented. Might initially existing restrictions improve spontaneously?
  • Why did so many patients drop out / did not reach at least 4 OTs? This should be explained in more detail
  • Possible disadvantages of osteopathic treatment are not discussed
  • Characteristics of the study participants are mainly lacking. Which maligniancies did they suffer from? Who received which treatment and which late effects/ which treatment complications occured? A table for illustration is recommended.

  • Column 44: This sentence does not make sense:....there are nearly 500,000 European children, adolescents and young adults with haematological malignancies (CAYA) survivors....  
  • Column 69: OE can support also the pre-clinical diagnosis of diseases characteristic of CAYA clinical history and 70 treatment and emerging from organs (visceral function). --> reference????
  • Column 184: Why did the patients experience pain during abdominal palpation? Did the authors exclude possible underlying conditions?
  • Column 248: The OE of these CAYA helped to support pediatricians in the differential diagnosis between primary issues of the structural system (myofascial pain) vs possible diseases of reproductive organs, bowel syndromes or urinary problems --> how did they support diagnosing this? Should be explained better as, in my opinion, this is not common knowledge among physicians

Author Response

(The authors gave the same response as above.)

Reviewer 4 Report

I read with interest the work titled "Osteopathic Treatment and Evaluation as Valuable Resource in the Clinical Setting of Childhood Haematological Malignancies." The paper is interesting, but remarkable there are methodological limitations.
Please, read the following suggestions constructively:
Major comments:
-  Please check the goals of the paper and the statistical analysis of the work. Statistical analysis should help to supply an answer to the questions of the study.  
- Please include inferential analysis ("pre" vs. "post" intervention) of your clinical assessment.
-  How did the authors evaluate the "safe" of the osteopathic intervention? In the first paragraph of the discussion, the authors indicated: "This study matched its aim to prove feasibility and safety of osteopathy in a paediatric haematologists-oncological cohort, including very frail patients, enrolled into a precision exercise-based training protocol." 

Minor comments:
- Introduction: Please, justify the relevance of physical activity/exercise as adjuvant treatment in cancer, especially in CAYA, in place of the relationship between fatigue and exercise. In this way, the introduction will be consistent with the goal of the paper.
- The study's design is based on an osteopathic intervention, but the authors define it as observational; why?
- Use the decimal point (".") consistently throughout the whole paper.

Author Response

(The authors gave the same response as above.)
